# Macrophages and Wnts in Tissue Injury and Repair

**DOI:** 10.3390/cells11223592

**Published:** 2022-11-14

**Authors:** Min Hee Park, Eun D. Lee, Wook-Jin Chae

**Affiliations:** 1Department of Microbiology and Immunology, Virginia Commonwealth University School of Medicine, Richmond, VA 23298, USA; 2Massey Cancer Center, Virginia Commonwealth University, Richmond, VA 23298, USA; 3Department of Obstetrics and Gynecology, Virginia Commonwealth University, Richmond, VA 23298, USA

**Keywords:** Wnt, macrophage, inflammation

## Abstract

Macrophages are important players in the immune system that sense various tissue challenges and trigger inflammation. Tissue injuries are followed by inflammation, which is tightly coordinated with tissue repair processes. Dysregulation of these processes leads to chronic inflammation or tissue fibrosis. Wnt ligands are present both in homeostatic and pathological conditions. However, their roles and mechanisms regulating inflammation and tissue repair are being investigated. Here we aim to provide an overview of overarching themes regarding Wnt and macrophages by reviewing the previous literature. We aim to gain future insights into how tissue inflammation, repair, regeneration, and fibrosis events are regulated by macrophages.

## 1. Introduction

### 1.1. Macrophages in the Immune System

Macrophages, one of the body’s most abundant populations of leukocytes, are mainly derived from the yolk sac during embryogenesis and are found in almost every tissue that plays an essential role during mammalian development [1,2]. They are specialized phagocytes, large vacuolated cells with abundant cytoplasm containing lysosomal granules [3]. Thus, they can be sentinels of the innate immune responses that serve to protect from the inflammatory process. In addition, they have important roles in tissue development, homeostasis, and remodeling [4,5,6]. Macrophages exhibit crucial homeostatic activity in nearly all body organs by producing growth factors and other mediators that provide trophic support to the tissues in which they reside.

Tissue-resident macrophages and monocytes recruited from the bone marrow are essential drivers of inflammatory and tissue regenerative responses that develop in response to tissue injury induced by infection, autoimmune disorders, mechanical or toxic injuries, and various other causes. These residents and recruited macrophage populations proliferate and undergo marked phenotypic and functional changes in response to growth factors and cytokines released in the local tissue microenvironment [6,7,8]. Recent studies have demonstrated these changes by identifying specialized and critically timed roles for different monocyte and macrophage activation states in tissue repair, regeneration, and fibrosis.

Typically, by simplified classification, these macrophages are primarily divided into two major groups, namely, classically activated macrophages (M1 macrophages) and alternatively activated macrophages (M2 macrophages) based on functions and expression patterns of genes and proteins [9,10,11]. The phenotype of M1 macrophages is activated by infection and pro-inflammatory Th1 cytokines, including bacterial lipopolysaccharide (LPS), interferon-gamma (IFN-γ), and tumor necrosis factor-α (TNF-α) [7,12]. M1 macrophages improve bactericidal capacity and increase pro-inflammatory cytokines such as IFN-γ, TNF-α, interleukin (IL)-1, IL-6, IL-12, and IL-23 [7,9,13]. In contrast, the M2 phenotype is induced by anti-inflammatory Th2 cytokine IL-4 and IL-13 and other factors [7,12]. M2 macrophages secrete anti-inflammatory cytokines, including IL-10, transforming growth factor-beta (TGF-β), and the cytosolic enzyme arginase-1 [7,9,13].

### 1.2. Wnt System

Wnt signaling is a conserved pathway across species. It is involved in various essential tasks by regulating cell differentiation, proliferation, stem cell development, immune cell functions, and tissue repair [14,15,16]. Evidence for the Wnt system’s pivotal role is that aberrant alterations of this molecular pathway are involved in multiple human disorders and pathologies, such as congenital abnormalities, autoimmune diseases, and cancer [17,18,19,20]. The Wnt pathways are initiated by the binding of a Wnt ligand to a receptor Frizzled (FZD) and a co-receptor such as low-density lipoprotein receptor-related protein 5/6 (LRP5/6) [21,22]. Wnt ligands and their receptors have multiple protein members; there are 19 different Wnt ligands, 10 FZD receptors, and diverse co-receptors such as LRP5/6 [20,21]. This molecular interaction between Wnt ligands and their receptors triggers the signaling cascade activation that activates or suppresses the expression of different genes, such as cyclin D1, Axin2, and Myc proto-oncogene [20]. Depending on the nature of the ligands and downstream events, Wnt signaling is broadly classified into canonical and non-canonical pathways: (1) the canonical signaling is activated via β-catenin, known as cadherin-associated protein-β, and members of the T cell factor (TCF)/lymphocyte enhancer-binding factor (LEF) family; and (2) the non-canonical pathway is independent of β-catenin and involves other components instead of TCF/LEF [23,24].

The binding of a Wnt1 class ligand (Wnt1, Wnt2, Wnt3, Wnt3a, Wnt8a, Wnt10a, Wnt10b, or Wnt16) generally induces the cascade of the Wnt signaling pathway (also called the Wnt/β-catenin pathway) via its binding to FZD receptor and low-density co-receptor LRP5/6 [24,25]. This signaling forms a transcriptional activation complex composed of β-catenin and TCF/LEF. The complex induces canonical Wnt target gene expression associated with cell proliferation, differentiation, and maturation [26,27]. In another Wnt pathway (previously known as the non-canonical Wnt pathway), a Wnt5a class ligand (Wnt4, Wnt5a, Wnt5b, Wnt6, Wnt7a, Wnt7b, or Wnt11) binds to its FZD receptor and another co-receptor including receptor-like tyrosine kinase (RYK), the receptor tyrosine kinase-like orphan receptor 2 (ROR2), tyrosine-protein kinase-like receptor 7 (PTK7), or the neurotrophin receptor homolog 1 (NRH1) [24,25]. Two pathways were addressed most so far: the Wnt/planar cell polarity (Wnt/PCP) and Wnt/calcium (Wnt/Ca^2+^) pathway [24,28]. The Wnt/PCP pathway begins when the Wnt ligand is recognized by the FZD receptor and the RYK/ROR2 complex. Wnt/PCP signaling activates Rho-associated kinase (ROCK) and c-Jun N-terminal kinase (JNK) to induce gene expression related to cell polarity, migration, and cytoskeletal arrangement changes. The Wnt/Ca^2+^ pathway begins with the FZD receptor being involved in the activation of calcium-dependent processes. The summary of the canonical and non-canonical Wnt signaling pathways is shown in Figure 1.

Notably, the categorization of Wnt pathways above is undergoing revisions with new findings. For example, Wnt2, which is typically elucidated as the canonical Wnt, can also activate the non-canonical Wnt pathway depending on the type of cells or tissues. There are results showing that Wnt2 plays an important role in cardiac formation and differentiation from embryonic stem (ES) cells via non-canonical Wnt signaling [29]. It is known that Wnt4, which was thought to trigger the non-canonical Wnt pathway, activates the canonical Wnt signaling pathway during myogenic differentiation [30]. Therefore, more investigations are warranted in the future to address signaling pathways with a given Wnt ligand with its receptors, co-receptors, and recipient cell or tissue types.

In addition, there were more investigations concerning soluble Wnt inhibitors, including Dickkopf (DKK) family members, sFRPs, and WIF1 [23,27,31].

In recent years, many researchers have focused on identifying and characterizing the different populations of macrophages that control the different stages of tissue repair, regeneration, and development in most organ systems [32]. Recent studies have suggested that various monocyte and macrophage populations play distinct and essential roles in chronic inflammation, tissue repair, regeneration, cancer, and fibrosis, as shown in Figure 2. Here, we discuss recent findings that have improved our understanding of the relationship and role of Wnt signaling and macrophages in tissue injury, repair, and regeneration. In particular, we highlight insights into Wnt and macrophages in the most immunologically active lung, liver, intestine, kidney, heart, and skin.

## 2. Macrophages and Wnt Signaling in Lung Injury and Repair

At least two types of macrophage populations are located in the lung. Alveolar macrophages (AMs) are located at the interface between the lung mucosa and the external environment. [33,34,35]. They play the role of primary defense by directly sensing immunological stimuli such as inhaled particulate elements and bacteria [36]. Other macrophages, called interstitial macrophages (IMs), inhabit the lung interstitium between the alveoli and the capillaries [34,35]. They come in direct contact with the matrix and other pulmonary connective tissue components and can phagocytose particles and bacteria [37]. In other words, IMs can serve as a secondary defense against the invasion of particles and bacteria evading phagocytic activity by AMs. Furthermore, IMs, which have unique transcriptional features, can be distinguished from AMs by their distinct surface phenotype [37,38].

Once lung injury occurs, mechanisms for regeneration are initiated to restore the lung epithelium. Wnt signaling is known to be essential for lung regeneration [39]. β-catenin, the main component of the canonical Wnt signaling, mediates pulmonary regeneration, acting as a transcription factor that stimulates the gene expression associated with epithelial regeneration and controlling the tight junctions of the epithelial cells in the lung. However, the investigation of the role of macrophage-derived Wnt ligands in the regeneration of lungs is in its infancy, and further studies are needed.

Lung macrophages are associated with interstitial lung diseases, such as idiopathic pulmonary fibrosis (IPF), which causes lung scarring for unknown reasons [40]. Hou and colleagues showed that Wnt/β-catenin signaling in M2 macrophages with significantly increased Wnt7a protein was activated, promoting differentiation of myofibroblasts by lung resident mesenchymal stem cells and exacerbating pulmonary fibrosis in mice [41]. In particular, they found that macrophages recruited into the fibrotic lungs of mice treated with bleomycin were mainly M2 macrophages. Wnt/β-catenin signaling activation in lung macrophages promoted fibrosis after bleomycin treatment [41,42,43]. Sennello et al. showed that lack of LRP5, the Wnt co-receptor, resulted in many fewer Siglec F*^low^* AMs, a macrophage cell type that caused pulmonary fibrosis [43]. Given that Wnt/β-catenin signaling affects lung macrophages contributing to the development and persistence of pulmonary fibrosis, targeting macrophages with activated Wnt/β-catenin signaling could lead to new strategies to slow lung fibrosis.

The importance of the Wnt pathway in lung macrophages was observed in infection and inflammatory processes. For example, it was confirmed that Wnt1, Wnt6, and Wnt10a were induced in an inflammatory environment such as the lung of Mycobacterium tuberculosis-infected mice, and especially, Wnt6 was a novel factor inducing macrophage polarization with an M2-like phenotype [44]. In addition, Zhou and colleagues [45] investigated the effect of the Wnt signaling regulator Rspondin3 on resolving inflammatory injury. They found that lung endothelial cells release Rspondin3 in response to inflammatory damage, activating Wnt/β-catenin signaling in the lung IMs. The specific deletion of Rspondin3 in endothelial cells prevented the production of anti-inflammatory IMs in endotoxemic mice and caused a severe inflammatory injury. In a study about cigarette smoke extract (CSE)-stimulated lung, non-canonical and pro-inflammatory Wnt5a was up-regulated by cigarette smoking, which induces parallel up-regulation of pro-inflammatory cytokines in mouse and human models [46]. In this study, it was confirmed that Wnt5a is a pro-inflammatory Wnt ligand and influences the polarization of the M1/M2 macrophage. When macrophages are activated following the activation of the Wnt5a pathway, they contribute to the inflammatory response of the lung. Zhu et al. [47] have shown that inflammation can be alleviated by inhibiting Wnt5a/JNK1-induced macrophage activation, which may be a target for treating chronic obstructive pulmonary disease (COPD). In patients with COPD, Wnt/β-catenin signaling was activated, resulting in increased alveolar epithelial cell marker expression, altered macrophage activity, and elastin remodeling [48]. Devi and Moharana [49] identified that infiltration and polarization of macrophage populations in the alveolar space of the COPD rodent model exposed to CSE trigger the neoplastic change and tumor growth via IL-6 mediated through activation of Wnt3a/β-catenin signaling cascade.

In various cancers, the Wnt signaling pathway plays an important role in the activity of tumor cells, promoting cancer metastasis and progression [50]. An abnormally increased activation of either the canonical or non-canonical Wnt signaling pathway was detected in lung cancer. In addition, circulating monocytes are changed into tumor-associated macrophages (TAMs) when recruited into the tumor microenvironments. Recently, it was reported that TAMs induce immune suppression, affecting lung tumorigenesis and development. Sarode and colleagues [51] provided strong evidence that β-catenin-mediated transcription plays the main role in the transition from tumor-inhibiting M1-like TAMs to tumor-promoting M2-like TAMs. Thus, targeting β-catenin in TAMs can furnish novel immunotherapy that reactivates antitumor immunity in the lung microenvironment.

## 3. Macrophages and Wnt Signaling in Liver Injury and Repair

Kupffer cells, also known as Kupffer–Borowicz cells, are the primary macrophages of the liver [52]. Kupffer cells are located in the liver sinusoid and are highly specialized for their phagocytic activity. They can sense danger-associated-molecular patterns (DAMPs) and pattern-associated-molecular patterns (PAMPs) via various receptors such as TLRs and Nod-like receptors (NLRs) [53].

Macrophages in the liver are one of the first responders to liver injury and are involved in modulating the fibrogenic response through several mechanisms. In addition, macrophages are closely related to the activated hepatic progenitor cells (HPCs) that occur parallel to fibrosis. Irvine and colleagues [54] investigated the role of Wnts derived from macrophages in chronic liver diseases (CLDs), especially concerning the HPC niche. Their results highlight that macrophage-derived Wnts have anti-fibrotic potential in CLDs and may be targeted for medical treatment. Carpino et al. [55] identified that the activated macrophages in pediatric nonalcoholic fatty liver disease (NAFLD) are closely associated with the HPC response through Wnt3a signaling. The study has shown that pro-inflammatory macrophages are the predominant subset of pediatric NAFLD and the important role of macrophage polarization in the progression of pediatric NAFLD.

In response to liver injury, quiescent hepatic stellate cells (HSCs) undergo a distinctive morphological transformation into proliferative, contractile, and extracellular matrix protein-producing myofibroblasts, leading to liver fibrosis. Akcora and colleagues [56] investigated the association between canonical Wnt signaling in HSCs and liver fibrogenesis using β-catenin/CBP inhibitor ICG-001. Interestingly, ICG-001 remarkably decreased collagen accumulation and HSC activation and significantly inhibited macrophage infiltration, intrahepatic inflammation, and angiogenesis. Therefore, it is suggested that inhibiting the canonical Wnt pathway can ameliorate liver fibrosis in vivo. To clarify the role of macrophage-derived Wnt ligands in regulating hepatobiliary injury and repair, Jiang and colleagues [57] investigated the effect of macrophage-specific deletion of Wntless, a cargo protein critical for cellular Wnt secretion. This study showed that a shortage of Wnt secretion in macrophages caused more hepatic injury induced by 3,5-diethoxycarbonyl-1,4-dihydrocollidine because of damaged hepatocyte proliferation and increased M1 macrophages, which accelerate immune-mediated cell injury.

The correlation between the Wnt signaling and liver macrophages was also observed in infection and inflammatory processes. For example, the overexpression of liver kinase B1 mediated mycobacterial infection in macrophages via FOXO1/Wnt5a signaling was identified [58]. Furthermore, it was suggested that the expression of LRP1 in macrophages promoted hepatic inflammation by controlling Wnt signaling [59].

TAMs are a major element of the tumor microenvironment and play a central role in the progression of hepatocellular carcinoma. A study has also shown that cancer-cell-derived Wnt proteins stimulate M2-like polarization of TAMs through the canonical Wnt/β-catenin pathway, resulting in growth, migration, metastasis, and immune suppression of cancer in hepatocellular carcinoma [60]. Obesity can stimulate the risk of tumor formation, and steatosis in the liver often leads to carcinogenesis. To determine the mechanism by which steatosis promotes cancer formation, Debebe and colleagues [61] used various liver cancer models in order to investigate the role of obesity in cancer. They showed that a high-fat diet lipid accumulation could activate Wnt/β-catenin signals, and pharmacological inhibition or loss of these signals suppress the growth of tumor-initiating cells (TICs) in vitro and reduce the accumulation of TICs in vivo. Their data also confirmed that Wnt/β-catenin, caused by steatosis-induced macrophage infiltration, promotes tumor progenitor cell growth.

## 4. Macrophages and Wnt Signaling in Intestine Injury and Repair

Macrophages in the intestine have roles in tissue homeostasis and inflammation, especially in the resolution of inflammation [34,62,63]. In recent years, Wnt signaling has played an essential role in intestinal epithelial proliferation and differentiation, and the expression of Wnt ligands by macrophages has been studied [39].

Saha and colleagues [64] analyzed the role of macrophage-derived Wnts in intestinal repair and regeneration after radiation injury in mice. Using macrophage-specific deletion of the Porcupine gene to inhibit Wnt ligand release in mice (*Csf1r.iCre-Porc*^flox/flox^), they showed that macrophage-derived Wnts contained in extracellular vesicles (EV) are important to mediate radiation-induced gastrointestinal syndrome (RIGS). Treatment of Wnt-containing EVs by ultracentrifugation of cell-free supernatant from bone marrow macrophages or using a total exosome isolation kit on irradiated mice facilitated the recovery of the irradiated mice. Cosín-Roger and colleagues [65] investigated the macrophage phenotype that determines Wnt ligands, the effect of macrophage phenotype on epithelial activation of Wnt signaling, and its relevance to the Wnt signaling pathway in ulcerative colitis (UC). They showed that M2 macrophages, not M1, activated Wnt signaling via Wnt1, which reduces the differentiation of enterocytes. In addition, the number of CD206-positive-M2 macrophages in the mucosa of UC patients significantly increased and acted as a source of Wnt1, showing that excessive Wnt signaling in the intestinal epithelium was involved in the development of colorectal adenocarcinoma. Other researchers [66] found that signal transducer and activator of transcription 6 (STAT6) mediates M2 polarization and induces the expression of Wnt2b, Wnt7b, and Wnt10a in the mucosa of 2,4,6-trinitrobenzene sulfonic acid-treated mice. Furthermore, they suggested that the STAT6-dependent macrophage phenotype activates the Wnt signaling pathway, promoting mucosal repair.

In another study [67], the number of CD206-positive cells, anti-inflammatory M2 macrophages, was significantly higher in colorectal cancer, whereas pro-inflammatory M1 macrophages were remarkably lower. In particular, the authors of this study investigated whether gastrins synthesized by colon tumor cells affect a pattern of macrophage infiltration in colon cancer. Interestingly, these results suggested that the expression of Wnt ligands was decreased in macrophages differentiated in the presence of progestin; it inhibited the acquisition of the M2 polarization in human macrophages.

## 5. Macrophages and Wnt Signaling in Kidney Injury and Repair

Macrophages are well known to increase in the diseased kidney and play a central role in kidney damage, inflammation, and fibrosis [68,69]. They exhibit a distinct phenotype with functional properties in response to various stimuli of the local microenvironment during injury, inflammation, fibrosis, and repair [70,71].

Lin and colleagues [72] investigated whether the canonical Wnt signaling pathway was activated during injury and played an essential role in repair in the kidney using mice subjected to kidney-ischemia-reperfusion injury. Their data showed that the Wnt7b produced by macrophages stimulated kidney repair and regeneration. Thus, it was suggested that renal macrophages could establish a beneficial kidney repair and regeneration system. Although several studies have demonstrated that kidney mononuclear phagocytes (MPs) are required for post-injury healing, they were not designed to identify a subpopulation of kidney MPs defined by phenotype. In addition, it has yet to be revealed whether kidney-resident macrophage (KRM) could potentially play a therapeutic role after acute kidney injury (AKI). In 2019, Lever and colleagues [73] found evidence that KRMs generate and respond to Wnt ligands and activate canonical Wnt signaling. They concluded that the regenerative source of KRMs after AKI is primarily in situ renewal as opposed to the infiltration of macrophage precursors in the blood and that KRM triggers the MHCII phenotypic transformation during development and after injury. After kidney injury, KRM was also rich in the Wnt signaling pathway, demonstrating that the pathways essential for mouse and human kidney development are activated. Their data showed that the mechanisms involved in kidney development in KRM might function after injury.

In recent years, many studies have investigated the role of Wnt/β-catenin in regulating macrophage activation and its contribution to renal fibrosis. Aberrant activation of the Wnt/β-catenin pathway is associated with renal fibrosis. Feng and colleagues demonstrated that Wnt3a enhanced M2 macrophage polarization induced by IL-4 or TGFβ1 caused STAT3 phosphorylation and nuclear translocation in vitro [74]. They also showed that β-catenin deletion of macrophages in the mice model attenuated fibrosis, macrophage accumulation, and M2 polarization observed in the kidney [75]. Thus, these results show that activation of Wnt/β-catenin signaling is essential to stimulate macrophage M2 polarization and promote macrophage proliferation during renal fibrosis. In another study, Feng et al. investigated how impaired regulation of the Wnt5a signaling in macrophages leads to renal fibrosis. In a mice model of kidney fibrosis, short hairpin RNA-mediated knockdown of Wnt5a expression reduced renal fibrosis and macrophage M2 polarization [76]. Their results showed that Wnt5a stimulates macrophage M2 polarization to promote renal fibrosis. Therefore, targeting Wnt signaling in macrophages may describe a new therapeutic strategy for protecting against renal fibrosis in patients with chronic kidney disease.

## 6. Macrophages and Wnt Signaling in Heart Injury and Repair

Macrophages and Wnt ligands are independently associated with cardiac development, reaction to cardiac injury, and repair [77]. Furthermore, Wnt signaling functions diversely in cardiovascular development and disease processes [78]. Monocytes and monocyte-derived macrophages are known to play important roles in the development of atherosclerosis and coronary heart disease, as well as in the immune response against cardiac ischemia [79,80].

After the heart is damaged, Wnt signaling is reactivated. There is increasing evidence that reactivation of the canonical Wnt signaling negatively affects infarct healing associated with cardiomyocyte death and cardiac fibrosis [79]. However, the effect of regulating the non-canonical Wnt signaling pathway in myocardial healing was not studied extensively.

Palevski and colleagues [81] investigated the role of macrophage-derived Wnt in the repair of myocardial infarction (MI). Their findings showed that the Wnt signaling pathway was activated after MI in mice and that macrophages expressed distinct components of the Wnt pathway and were a source of non-canonical Wnt after MI. In addition, they revealed that inhibition of macrophage Wnt5a secretion could block the inflammatory autocrine loop and convert macrophages to the M2-like phenotype. These M2 macrophages reduced excessive inflammation and enhanced infarct repair. Meyer and colleagues [82] studied microenvironment-dependent changes in inflammatory monocytes after MI for activation of monocytes, which play an essential role in healing after MI. They found more components of the non-canonical Wnt pathway and more inhibitors of the intracellular canonical Wnt pathway in the monocytes isolated from the heart than in the bone marrow. It was also revealed that cardiomyocytes constitute a significant source of Wnt inhibitory factor 1 (WIF1) after MI. In this study, WIF1 interferes with the non-canonical Wnt signaling pathway of monocytes and reduces their pro-inflammatory signaling activation. Overall, Wnt signaling of macrophages is related to cardiac remodeling after MI, and macrophage-derived Wnt may be a new therapeutic target to improve infarct healing and recovery.

## 7. Macrophages and Wnt Signaling in Skin Injury and Repair

Macrophages are well known to play essential roles and coordinate in all stages of the skin wound healing process [83,84,85].

A skin injury can provide an ideal model for studying the role of the innate immune system between regeneration and fibrotic healing. Recently, the wound-induced hair neogenesis (WIHN) model, which can induce fibrotic scarring, was used to investigate the potential role of macrophages in determining healing fate by Gay and colleagues [86]. Their results showed that late wound macrophages phagocytosed the dermal Wnt inhibitor SFRP4 to establish sustained Wnt activity, leading to fibrosis. In addition, the phagocytosis of SFRP4 by macrophages in the human hidradenitis suppurativa was related to the recovery of fibrotic skin. These results revealed that macrophages could change the fate of skin wound healing by regulating major signaling pathways via phagocytosis.

Macrophages are known to regulate developmental vascularization through non-canonical Wnt signaling and are associated with wound angiogenesis. Stefater III and colleagues [87] showed that wound macrophages use the Wnt-Flt1 signaling pathway via Flt1, a receptor for vascular endothelial growth factor A. Calcineurin is an important mediator in regulating wound response. Thus, they found that macrophages use Wnt-Calcineurin-Flt1 signaling to inhibit angiogenesis and slow repair.

To investigate the effect of perifollicular macrophage-derived Wnt on the activation of hair follicle stem cells (HF-SCs) and the induction of anagen (the active growth phase of hair follicles) in the hair cycle in mice, Castellana and colleagues [88] used and injected subcutaneously into mice a liposome containing IWP-2, a specific hydrophobic small molecule inhibitor of Wnt. Taken together, their results suggest that the apoptosis-related secretion of Wnt by skin-resident macrophages contributes to the activation of HF-SCs, allowing HF to enter the anagen growth phase of the hair growth cycle. Based on this, the function of macrophages in human skin was recently studied. As a result, similar to murine perifollicular macrophages, human macrophages expressed many Wnt10a and Wnt7b proteins during anagen. The proteins significantly decreased during catagen (a short transitional phase in the hair growth cycle) [89]. Therefore, it is found that perifollicular macrophages are worthy of attention as a therapeutic target for skin repair, inflammatory skin disease, and cancer.

## 8. Future Perspective and Important Questions to Ask

Wnt ligands were studied extensively in developmental and cancer biology due to their important roles in cell differentiation and proliferation. However, it is still unclear how Wnt ligands regulate immune cells, including macrophages, in various inflammatory diseases. We summarized the previous findings in Table 1.

Over the years, it has become increasingly clear that tissue injury, repair, and remodeling are fundamental biological processes to maintain homeostasis. The recent development of the genomics approach, including single-cell RNA (scRNA) sequencing, enables us to ask important questions.

First, what are the mechanisms of Wnt ligands to regulate macrophages in various human diseases? Our current understanding of the topic needs to be revised to provide a comprehensive picture of how Wnt ligands regulate tissue injury and repair processes. The dynamics of Wnt ligands’ expression and secretion is an important study point. As covered in this review, macrophages are regulated by Wnt ligands but are also sources of Wnt ligands. Cellular and molecular mechanisms and their implications in different tissues and organs need to be delineated in the future.

Second, how does a subset of macrophages “sense” Wnt ligands? Since macrophages are heterogenous populations with various origins, dissecting diverse macrophage populations and identifying subsets of macrophages regulated by Wnt ligands will be important for understanding tissue injury and repair. The source of Wnt ligands and macrophages as their target cells will provide valuable insights regarding cellular crosstalk for tissue injury and repair.

## Figures and Tables

**Figure 1 cells-11-03592-f001:**
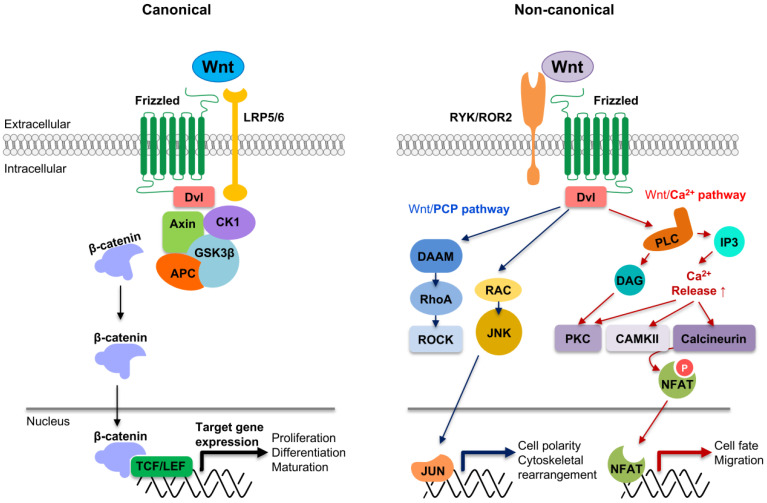
The canonical and non-canonical Wnt signaling pathways. In the canonical Wnt pathway, the Wnt signaling is activated upon binding Wnt ligands such as Wnt3a to Frizzled (FZD) and co-receptor LRP5/6. Then, the Disheveled (Dvl) recruits axis inhibition protein (Axin), the casein kinase 1 (CK1), and glycogen synthase kinase 3 β (GSK3β) to the plasma membrane, inactivating the β-catenin destruction complex and weakening phosphorylation and degradation of β-catenin. This results in the accumulation of the stabilized β-catenin in the cytoplasm and the translocation of it into the nucleus. β-catenin in the nucleus forms an active transcriptional complex with T-cell factor (TCF) and lymphoid enhancer factor (LEF), leading to canonical Wnt target gene expression. Upon the non-canonical Wnt ligands such as Wnt5a binding to the RYK/ROR2-FZD complex, Dvl is recruited, and Wnt/PCP or Wnt/Ca^2+^ signaling pathway is activated. In the Wnt/PCP pathway, the scaffold protein Dvl stimulates the activation of the small GTPase Rho and RAC to induce Rho-associated kinase (ROCK) and c-Jun N-terminal kinase (JNK), respectively. ROCK and JNK trigger gene expression associated with cell polarization and cytoskeletal rearrangement. In the Wnt/Ca^2+^ pathway, Dvl activates phospholipase C (PLC), stimulating 1,2-diacylglycerol (DAG) and inositol 1,4,5-triphosphate (IP3). The activated IP3 promotes the release of Ca^2+^ within the cytoplasm, and protein kinase C (PKC), CAMKII, and calcineurin are subsequently induced. The nuclear factor of activated T-cells (NFAT), a transcriptional factor, is then activated through dephosphorylation to induce calcium-dependent cytoskeletal and transcriptional responses.

**Figure 2 cells-11-03592-f002:**
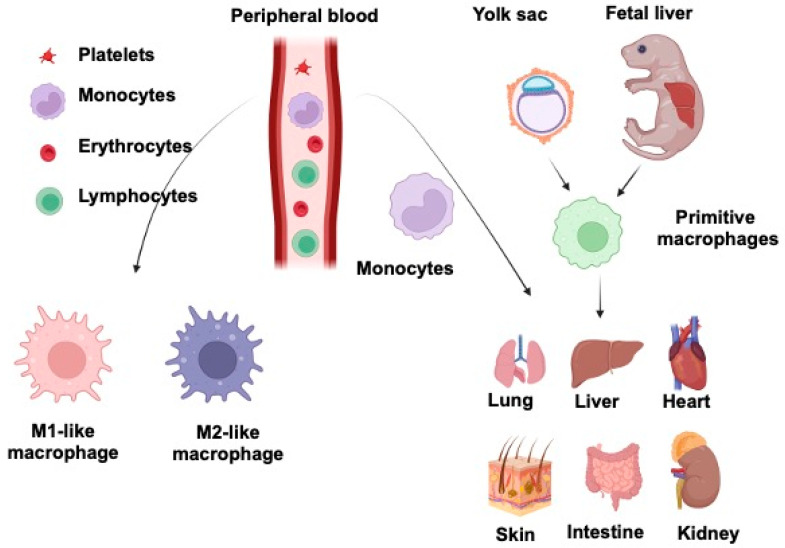
Macrophage development and functions in different organs. Tissue-resident macrophages are originated from the yolk sac/fetal liver and hematopoietic progenitors/circulating monocytes. Monocytes are further differentiated into M1-like and M2-like macrophages based on their expression markers upon a variety of stimuli. M2-like macrophages are known to be important for wound repair and also share similar features with TAMs. M1-like macrophages constitute the first line of defense against intracellular pathogens. Macrophages are sources of Wnt ligands and mediate Wnt ligand-mediated signaling for various immune responses for tissue inflammation and repair. This was created with Biorender.com.

**Table 1 cells-11-03592-t001:** Summary of study results related to macrophages and Wnts in tissue injury and repair.

Organ/Tissues	Tissue-Resident Macrophages/Cells	Injuries/Diseases	Macrophages and Wnts	References
Lung	Alveolar macrophages	Pulmonary fibrosis	Activation of Wnt/β-catenin signaling in alveolar macrophages leading to disruption of repair and promotion of fibrosis in lung	[41,42,43]
		Mycobacterial infection	Wnt6 causing macrophage polarization with M2-like phenotypes	[44]
		Inflammatory injury	Exacerbation of inflammatory injury due to inhibition of anti-inflammatory interstitial macrophage, Influence of a pro-inflammatory Wnt5a ligand on M1/M2 macrophage polarization	[45,46]
		Chronic obstructive pulmonary disease	Increased contribution of macrophages on inflammatory response due to activation of Wnt5a/JNK1 pathway, Change in macrophage activity via activation of Wnt/β-catenin signaling	[47,48,49]
		Lung cancer	Transition to tumor-promoting M2-like tumor-associated macrophage due to Wnt/β-catenin-mediated transcriptional activation	[51]
Liver	Kupffer cells	Chronic liver disease (e.g., liver fibrosis)	The anti-fibrotic potential of Wnt derived from macrophages, Reduction in collagen accumulation and macrophage infiltration in inhibition of canonical Wnt pathway	[54,56]
		Pediatric nonalcoholic fatty liver disease (NAFLD)	Correlation of pro-inflammatory macrophage activation and hepatic progenitor cell response through Wnt3a pathway in NAFLD	[55]
		Partial hepatectomy	Wnt secretion from Kuffer cells for β-catenin activation for liver regeneration	[57]
		Mycobacterial infection	Control of mycobacterial infection in macrophage via FOXO1/Wnt5a signaling	[58]
		Hepatic inflammation	Promoted liver inflammation caused by modulation of Wnt signaling via LRP1 expression in macrophages	[59]
		Liver cancer	Stimulation of M2-like macrophage polarization through the canonical Wnt signaling of cancer cell-derived Wnt ligands, Promotion of tumor cell growth by Wnt/β-catenin signal induced by high-fat diet lipid accumulation and steatosis-induced macrophage infiltration	[60,61]
Intestine	Intestinal macrophages	Radiation injury	Macrophage-derived Wnts, an essential element for intestine regeneration	[64]
		Inflammatory bowel disease (IBD) (e.g., ulcerative colitis)	Activated Wnt signaling in epithelial cells caused by M2 macrophage through Wnt1, which impaired enterocyte differentiation, Promoting mucosal repair via the Wnt signaling pathway of STAT6-dependent macrophage	[65,66]
		Colorectal adenocarcinoma	Increased CD206-positive M2 macrophages and exaggerated Wnt signaling in colorectal cancer	[65,67]
Kidney	Renal macrophages	Kidney-ischemia-reperfusion injury	Stimulation of renal repair and regeneration of macrophage-derived Wnt7b	[72]
		Acute kidney injury	Wnt ligand generation and canonical Wnt signaling activity in macrophages after kidney injury	[73]
		Renal fibrosis	Stimulation of M2 macrophage polarization causing renal fibrosis due to increased Wnt signaling	[74,75,76]
Heart	Cardiac macrophages	Myocardial infarction (MI)	Macrophage as a source of non-canonical Wnt after MI Reduction in dramatic inflammation and improvement in the repair by M2 macrophage	[81,82]
Skin	Langerhans cells	Wound-induced hair neogenesis, Human hidradenitis suppurativa	Phagocytosis of macrophages on dermal Wnt inhibitor SFRP4	[86]
		Wound angiogenesis	Inhibition of angiogenesis and repair using Wnt-Calcineurin-Flt1 signaling in macrophages	[87]
		Hair growth	Increase in macrophage-derived Wnts in the hair growth cycle	[88,89]

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
