# Peer review of "Macrophages and Wnts in Tissue Injury and Repair"

_cells, 2022, doi:10.3390/cells11223592_

Round 1
Reviewer 1 Report
This is a useful review with a wide perspective on the interplay between Wnt signalling and Macrophage function in a number of adult tissues in response to tissue injury and their combined role in tissue repair and homeostasis.
There is probably a need in the future for more specific reviews focussing on specific organ systems and tissue types, but the overview approach that the authors have chosen here is certainly very useful at this stage; and their focus is on Macrophages, which is useful.
From my perspective the introduction to Wnt signalling sounded a but old-fashioned including some outdated concepts, such as 'Wnt1 class ligands' and 'Wnt5a class ligands' based on the original frog axis duplication and mouse mammary tumor virus experiments (when we do now know that Wnt4 can function in canonical (Wnt/beta-catenin) (e.g. in somites) and Wnt2 may activate the non-canonical pathway in mouse EScell differentiated into mesoderm). I would suggest just to tone down the language slightly rather than re-writing the whole paragraph. It is also now less than clear whether there are really just two clearly separable non-canonical pathways, again I would just tone down the language to express the remaining ambiguity a bit more accurately than in the current text.
Later in the review the role of WIF1 after MI and SFRP4 in the skin are described, I therefore wonder whether the authors would consider introducing this concept of secreted Wnt inhibitors already in this introductory subsection on Wnt signalling.
In section 6 on heart injury and repair, I very much agree with the end of the first paragraph, 'In the case of Wnt signalling, ...', however, since there are no citations to references, it is not clear whether the authors are citing others or whether they are expressing their own personal conclusions from the literature that they have studied. If it is not their own conclusions then the authors should cite relevant literature.
There is only one display item, Table 1, which is very useful. However, I wondered whether the authors might consider whether some of their readers would find it useful to have additional schematic figures outlining Wnt signalling, and Macrophage structure and function in the introduction, and possibly some illustration of the proposed roles in individual tissues.
Writing style and English was generally very good, in the Abstract I would probably write 'we aim to provide an overview' rather than 'we aim to overview'. In the Wnt system introduction, the authors do not really introduce the following family members (line 28) but the numbers of family members.
Author Response
We truly thank reviewer 1. Please see the attachment.

Reviewer 2 Report
As the titles indicates, this review article describes the role of Wnt and macropahges in tissue injury and repair. For the most part, the article is well organized and written, but I suggest the following changes:
Abstract (L13-16): The last sentence does not make sense and needs to be rewritten.
Introduction: because macrophages are the cells that Wnt is being examined in, I recommend you switch sections 1.1 and 1.2 so that the macrophage section appears first.
The Wnt pathways are so complicated that a figure is needed so that the reader does not need to pull up diagrams from other papers.
L25: rather than "disabilities" do you mean "abmormalities"?
L26-29: needs to be rewritten, doesn't make sense.
L30: what molecular interaction? Perhaps once the preceding sentence is corrected this will be obvious.
L97-98: doesn't make sense.
L103: change "phagocytosis" to "phagocytose"
L107: change to "Once lung injury occurs...."
L117: change "myofibroblast" to "myofibroblasts"
L122-124: rewrite, does not make sense.
L127-150: should it be "Rspondin3" or "R-spondin3"? Be consistent.
L164-165: this statement does not occur in this reference. Please double-check that all of your references are correctly numbered.
L196: define TAMS
L214-217: please rewrite this sentence to be more clear.
L217: "the fraction of extracellular vesicles" Fraction obtained how? Please describe.
L264: what do you mean by "aggravated"?
L271-274: please rewrite to make clearer
L305: change to "monocytes'"
L311-312: this sentence is redundant
L314: change "dealing with" to "studying"
L324: change "related to" to "associated with"
L337-338: explain what anagen and catagen are
L345: change to "We summarize previous findings in Table 1."
L346: change to "...it has become..."
L347-348: rewrite - makes no senes
L360: change to "...will be important to understanding..."
Table and References: be sure that you use the correct form (macrophage or macrophages) throughout the table.
References are only included in this section up to 86, however, you include referenced up to 91 in the table. Should all of these also be included/cited within the body of the manuscript as well?
Author Response
We truly thank reviewer 2. Please see the attachment.
